Exploration of the “larval pool”: development and ground-truthing of a larval transport model off leeward Hawai‘i

Wren Johanna L.K. 1 jwren@hawaii.edu
Kobayashi Donald R. 2
1 Department of Oceanography, School of Ocean and Earth Science and Technology (SOEST), University of Hawai‘i at Mānoa , Honolulu Hawai‘i , United States
2 NOAA Fisheries, Pacific Islands Fisheries Science Center , Honolulu Hawai‘i , United States
Esteban María Ángeles
Electronic publication date: 2016 Feb 1
Publication date: 2016
Volume: 4
Electronic Location ID: e1636
Received 2015 Jan 8; Accepted 2016 Jan 7
Copyright: ©2016 Wren and Kobayashi
Copyright year: 2016
Copyright holder: Wren and Kobayashi
License: This is an open access article distributed under the terms of the Creative Commons Attribution License, which permits unrestricted use, distribution, reproduction and adaptation in any medium and for any purpose provided that it is properly attributed. For attribution, the original author(s), title, publication source (PeerJ) and either DOI or URL of the article must be cited.
License URL: https://creativecommons.org/licenses/by/4.0/

Keywords: Connectivity, Larval dispersal, Larval pool, Recruitment, Settlement, Acanthurid, Hawaii

Funding: Kona Integrated Ecosystem Assessment Program of NOAA Fisheries Pacific Islands Fisheries Science Center HIMB-NWHI MOA-2009-039/7932 NSF OCE 12-60169 SeaGrant NA14OAR4170071 Funds for JLKW were provided by the Kona Integrated Ecosystem Assessment Program of NOAA Fisheries, Pacific Islands Fisheries Science Center, with additional support provided by the HIMB-NWHI partnership (MOA-2009-039/7932), NSF (OCE 12-60169) and SeaGrant (NA14OAR4170071). This is SOEST contribution number 9562. The funders had no role in study design, data collection and analysis, decision to publish, or preparation of the manuscript.

==============================
Most adult reef fish show site fidelity thus dispersal is limited to the mobile larval stage of the fish, and effective management of such species requires an understanding of the patterns of larval dispersal. In this study, we assess larval reef fish distributions in the waters west of the Big Island of Hawai‘i using both in situ and model data. Catches from Cobb midwater trawls off west Hawai‘i show that reef fish larvae are most numerous in offshore waters deeper than 3,000 m and consist largely of pre-settlement Pomacanthids, Acanthurids and Chaetodontids. Utilizing a Lagrangian larval dispersal model, we were able to replicate the observed shore fish distributions from the trawl data and we identified the 100 m depth strata as the most likely depth of occupancy. Additionally, our model showed that for larval shore fish with a pelagic larval duration longer than 40 days there was no significant change in settlement success in our model. By creating a general additive model (GAM) incorporating lunar phase and angle we were able to explain 67.5% of the variance between modeled and in situ Acanthurid abundances. We took steps towards creating a predictive larval distribution model that will greatly aid in understanding the spatiotemporal nature of the larval pool in west Hawai‘i, and the dispersal of larvae throughout the Hawaiian archipelago.

Introduction

Population connectivity, defined as the exchange of individuals among geographically separate sub-populations (Cowen et al., 2007; Fogarty & Botsford, 2007; Pineda, Hare & Sponaugle, 2007; Cowen & Sponaugle, 2009) is widely recognized as important in effective marine conservation (e.g., Jones, Srinivasan & Almany, 2007; Toonen et al., 2011; Treml et al., 2012). Because traditional maximum sustainable yield fisheries management has consistently failed (Sharp, 1995; Hilborn, 2011), there is a general movement towards Ecosystem Based Management (EBM) to replace single-species management approaches (Slocombe, 1993; Browman et al., 2004; Pikitch et al., 2004; Kappel, Martone & Duffy, 2006; McLeod & Leslie, 2009). A common approach in EBM is to establish networks of marine protected areas (MPAs) to protect breeding populations which can then “spill-over” into surrounding areas to enhance stocks in areas of heavy human exploitation (Gerber et al., 2005; Halpern, Lester & Kellner, 2009; Lester et al., 2013). Management efforts focus on regulating the adult fish, but an effective management for healthy ecosystems and sustainable fisheries requires understanding of not only adult habitats and spawning areas, but also larval dispersal and population connectivity (Gaines et al., 2010; Toonen et al., 2011).

The majority of marine organisms show a bi-phasic life style with a free-living, dispersive larval stage, that can last hours to weeks, and a more sessile or sedentary adult form (Thorson, 1950; Thresher & Brothers, 1985; Strathmann, 1993; Raff, 2008; Marshall et al., 2012). Although fish are mobile animals, most adult reef fish show strong site fidelity (Meyer et al., 2000; Meyer, Papastamatiou & Clark, 2010; Wetherbee et al., 2004), thus dispersal is primarily limited to the egg and larval stages when the pelagic propagules can be transported across wide geographic areas depending upon the oceanographic regime. The importance of the larval life stage of marine fish and invertebrates in understanding population dynamics has long been recognized (Thorson, 1950; Knight-Jones, 1953; Scheltema, 1971) yet little is still known about dispersal of marine larvae, and factors affecting dispersal and near shore retention (Levin, 2006; Hellberg, 2009; Cowen & Sponaugle, 2009). However, an important driver of horizontal dispersal is the vertical strata occupied by these early life-history stages (Olivar & Sabate, 1997; Muhling & Beckley, 2007) since abiotic and biotic gradient vary greater on the vertical scale than the horizontal scale.

The Hawaiian Archipelago is located in the central north Pacific gyre and ocean circulation patterns are primarily driven by prevailing easterly trade winds (Lumpkin, 1998). The main surface flow along the island chain is from east to west: from the Main Hawaiian Islands (MHI) to the North Western Hawaiian Islands (NWHI) and westward into the Central and Western Pacific (Fig. 1). The Hawaiian Lee Counter Current that flows along the southern perimeter of the islands is the main mode of surface flow towards the Hawaiian Island chain from the west to the east (Chavanne et al., 2002; Qiu & Durland, 2002). Cyclonic (cold core) and anti-cyclonic (warm core) wind-generated mesoscale eddies are consistently formed in the lee of the Hawaiian Islands (Patzert, 1969; Lumpkin, 1998; Jia et al., 2011). Cold core eddies travel northeast along the southern perimeter of the island chain, transporting water from the Island of Hawai‘i, henceforth referred to as Big Island, along the MHI chain (Seki et al., 2001). Warm core eddies are spun up off the Big Island and travel southwest, transporting water away from the island chain (Bidigare et al., 2003). These eddies have the potential to boost primary production as well as aid with water dispersal or retention within the islands themselves (Seki, Lumpkin & Flament, 2002).

Figure 1 Main surface currents around Hawaii.

Main surface currents around the Main Hawaiian Islands (after Lumpkin, 1998).

The hydrodynamic environment around the Hawaiian Islands is complex making it difficult to generalize physical dispersal pathways for pelagic propagules (Friedlander et al., 2009). It has long been hypothesized that the mesoscale eddies that form in the lee of the Hawaiian Islands are responsible for retaining larval fish near the islands, preventing the larvae from being swept away from appropriate settlement habitat (Lobel & Robinson, 1983; Lobel, 1989; Lobel, 2011). The majority of spawning in Hawaiian reef fishes takes place during spring and summer (February through June), although most reef fish are believed to have low levels of spawning year round (Watson & Leis, 1974; Lobel, 1989; Bushnell, Claisse & Laidley, 2010). Reef fish spawning season in Hawai‘i coincides with “eddy season” (Lobel & Robinson, 1983; Lobel, 1989; Lobel, 2011); that is, the time of year when eddies are more prevalent (Flament et al., 1996). Donahue et al. (2015) showed that timing of spawning and the subsequent timing with the presence of near shore eddies is more important than the location of spawning for larval settlement success in the lane snapper (Lutjanus synagris) off Cuba. In Hawai‘i, the temporal and spatial dynamics of eddy entrainment of propagules is not well understood, and while mesoscale eddies are shown to have an effect on larval dispersal elsewhere (Baums, Paris & Chérubin, 2006; Harrison, Siegel & Mitarai, 2013), the effects of mesoscale eddies in larval transport and settlement in Hawai‘i has proven contradictory (Vaz et al., 2013), ranging from shown correlations between time of spawning and eddy activity (Lobel, 1989) to no observed effect on recruitment (Fox et al., 2012). Spawning has also been linked with lunar cycles, and many reef fishes in Hawai‘i are shown to spawn on or near full moon (Walsh, 1987; Bushnell, Claisse & Laidley, 2010).

Studies on larval reef fish distributions suggest that larval abundances of insular marine fish that spawn pelagic eggs are more numerous off shore than near shore (see Leis & Miller, 1976, J Wren & D Kobayashi, pers. obs., 2011). Many of the most common reef fish species in Hawai‘i fall into this category, and larvae of Pomacentrids, Chaetodontids and Acanthurdis are very rare in inshore plankton tows, but numerous in deeper, offshore tows (Leis & Miller, 1976) although this does depend on the stages of the larvae sampled by the gear. Studying later stage larvae in situ is generally intractable, because most shorefish larvae are not easily captured/identified/enumerated, are found in low densities, and are spread over large geographical areas (Clarke, 1991). One approach that could yield insights into the vertical and horizontal distributions of these elusive early life-history stages is using computer simulation coupled with field sampling (Botsford et al., 2009; Leis, Herwerden & Patterson, 2011; Leis et al., 2013; Sponaugle et al., 2012; Kough, Paris & Butler, 2013; Wood et al., 2014).

Given the common spawning season and the presence of mesoscale eddies in the lee of the Hawaiian Islands, we wanted to test the hypothesis that there is a common “larval pool” in the lee of Big Island (Sale, 1978). This was accomplished with field sampling and through numerical modeling. We use two different biophysical modeling approaches to investigate larval distributions. The first approach uses a computer simulation to hindcast likely sites of propagule origin from midwater trawl samples and employs statistical tools to indicate the most likely vertical strata of propagule occupancy. This information will improve understanding of horizontal and vertical movements as well as be useful for targeted sampling of these elusive early life history stages. The second approach uses computer simulation to forecast field samples, in essence generating a virtual “larval pool,” and uses a similar comparative approach to identify the most likely vertical strata of propagule occupancy using observed and predicted patterns of relative larval abundance.

Methods

Field sampling

Midwater trawl samples collected during the SE1104 research cruise aboard the NOAA ship Oscar Elton Sette July 1-July 14, 2011 were used in this analysis. Two replicate 2-hour dual-warp Cobb midwater trawls (see Baltz et al., 2002 for gear specifications) were conducted nightly, at ca. 2100 and 0100 h, at a speed of 3 knots at 11 stations off the west coast of the Big Island (Fig. 2). Maximum trawl depth targeted the shallow scattering layer, and was determined nightly from EK60 active acoustics identification of the lower boundary of the shallow scattering layer, which ranged between 150–250 m. Each trawl sampled three depth strata evenly (stepped oblique) divided between 50 m and the deepest bounds of the scattering layer. Tow time for each depth interval was 20 min following net equilibration at the deepest depth, for a minimum duration of 1h at depth and a total trawl time of approximately 2h. The trawl is constructed of graded nylon mesh (6′, 3″, and a 1.5″ liner terminated with a 1 mm Nitex plankton net) and has a mouth area of approximately 690 m2 at the most forward position of the coarsest mesh.

Figure 2 Location of trawl sites off Big Island of Hawai‘i.

Location of Cobb trawl stations off the Kona coast of Hawai‘i Island used in this stud and visited by the SE1104 cruise.

Trawl catch was quantified based on weight and volume, and the catch was sorted into six categories: Myctophid fish, non-Myctophid fish, cephalopods, crustaceans, gelatinous zooplankton and miscellaneous zooplankton. From the non-Myctophid fish category, shore fish larvae were removed and preserved in 95% EtOH. Fish larvae from seven familial or similar groups, Chaetodontidae; Acanthuridae; Pomacanthidae; Labridae and Cirrhitidae; Muraenidae; Aulostomidae and Fistulariidae; and Mullidae, were measured and counted. These groupings were chosen because they are relatively easy to distinguish visually and their members are almost exclusively shore fish (Mundy, 2005; Randall, 2007). Identification to species level was not possible in the scope of this study due to time constraints and difficulty/impossibility of identification without genetic analysis (B Mundy, pers. comm., 2011) for many larval fish (e.g., Acanthurids, Pomacanthids). It is important to note that we used trawl catch data for the family of Acanthurids, not specifically for Zebrasoma flavescens.

Dispersal modeling

Larval dispersal simulations were performed using a two dimensional, Lagrangian advection diffusion model (Polovina, Kleiber & Kobayashi, 1999; DeMartini, Wren & Kobayashi, 2013). In the model, larvae are moved at each timestep by advective displacement caused by water flow combined with a random displacement caused by diffusion. Current velocities used by the model consist of daily snapshots of u (east) and v (north) velocities obtained from a regional implementation of the HYbrid Coordinate Ocean Model (HYCOM) for the Main Hawaiian Islands (MHI) (HYCOM described in the following section). We used an eddy diffusivity of 250m2∕s, which is derived from drifters in the MHI (Y Jia, pers. comm., 2012, Rivera et al., 2011). Spawning habitat was defined in the model as a suite of 302 sites uniformly located every 5 km around the perimeter of each of the emergent MHI. The dispersal model is conventionally run forward in time, but is capable of running in reverse (backwards in time) for hindcasting purposes, facilitated by using off-line storage arrays of the daily currents.

Ocean circulation data

The HYbrid Coordinate Ocean Model (HYCOM), a data-assimilative ocean circulation model developed by the multi-institution HYCOM Consortium, was the source of the modeled ocean circulation flow fields used in this study. Forcing and assimilation for HYCOM are accomplished using the Navy Operational Global Atmospheric Prediction System (NOGAPS) winds to drive the dynamics, and the Navy Coupled Ocean Data Assimilation (NCODA) system to incorporate external measurements of satellite altimetry, sea surface temperature, and in-situ vertical profiles of temperature and salinity. The NOGAPS and NCODA data can vary in temporal resolution from hours to weeks as well as be snapshot inputs from vessels, moorings, or profiling buoys; however, the standard temporal resolution of the HYCOM output data is daily, which is what was used in this study (Jia et al., 2011). The daily HYCOM data is a regional implementation of the University of Miami global model using K-Profile Parameterization for characterizing the mixed layer, and is made available at the Asia-Pacific Data-Research Center (APDRC) at the International Pacific Research Center (IPRC), School of Ocean and Earth Science and Technology (SOEST), University of Hawai‘i (point of contact Dr. Yanli Jia). Daily HYCOM data covering the temporal domain of the calendar year 2011 and the spatial domain of 166°W to 150°W longitude and 16°N to 26°N latitude were downloaded as NetCDF files using the APDRC/IPRC website as a data conduit.

HYCOM is vertically structured with 32 layers ranging in thickness from 5–500 m, with thinner layers in the upper portions of the water column and an average layer width of 10 m in the upper 40 m of the water column. For this exercise, several different vertical strata and averages of vertical strata of the HYCOM data were used: surface waters representing the upper 5 m only, 50 m depth only, 100 m depth only, 0–50 m average, and 0–100 m average. Extraction and averaging was undertaken using R language subroutines (R Core Team, 2012). The horizontal structure of the HYCOM is a variable width resolution averaging ∼0.04 degree over both longitude and latitude. For ease of use, the daily variable resolution output was regridded to a daily uniform 0.04 degree coordinate system in latitude and longitude for the study domain using the mapping software Generic Mapping tools (GMT) subroutines named blockmean and surface (Wessel & Smith, 1991).

Hindcast simulation

Hindcasting simulations used the dispersal model (described above) run backwards in time with reversed currents (i.e., starting at the trawl site and extending up to 75 days prior to the trawl date). This approach was used to predict the likely origination site and depth strata of occupancy for larval shore fish caught at the 11 trawl sites by comparing the five different depth scenarios (depths = the upper 5 m, 50 m depth, 100 m depth, 0–50 m average, and 0–100 m average) and by using a plausible range of pelagic larval durations (PLDs = 15, 30, 45, 60, and 75 days). The averaged depth strata were used in an effort to emulate diel vertical migration (DVM) over 50 m and 100 m vertical horizons, realizing that this is only an approximation of what DVM would cause. For each trawl sample and PLD/depth combination, 1,000 simulated hindcast trajectories were generated for analysis. When a particle was within 5 km of the 302 pre-defined spawning sites on day 1 it was defined as originating from that spawning area.

Forecast simulation

In the forecast simulation the dispersal model was run in conventional forward mode. This approach was intended to simulate a generic “larval pool” for shorefish (sensuSale, 1978), and specifically for the yellow tang, Zebrasoma flavescens. We used the HYCOM current vectors from the 100 m depth layer, based on results from our hindcast simulation (see results section) and the PLD was parameterized after the yellow tang and set to 54 days (Claisse, McTee & Parrish, 2009; Christie et al., 2010). Size at settlement for Z. flavescens is estimated at 30–55 mm total length (Claisse, McTee & Parrish, 2009). Virtual larvae were released daily from the spawning habitat during 2011. For each of the 302 release sites and for each day, 1,000 simulated forecast trajectories were generated for analysis. Pelagic abundance of older larvae (defined as ages ≥50% of PLD) was tabulated into daily 0.1° × 0.1°bins for the trawl survey dates, and actual larval catch of Acanthurids was compared to simulated larval abundance at that point in time and space. There are to our knowledge no published growth rates for larval yellow tang, thus the age of the Acanthurids used to parameterize our model are estimates. Based on the size at settlement (∼30 mm) and a 55 day PLD, we felt comfortable comparing trawl catch with data modeled using a 54 day PLD and estimating the larval pool using 50% of the PLD.

Statistical methods

A generalized additive model (GAM) was constructed in R using the mgcv package (Wood, 2011), and used to characterize effects from vertical layering and PLD on the predicted origin. Operationally this was accomplished by using the GAM to predict hindcast settlement success. Smoother functions for the significant terms were plotted and examined. Degrees of freedom for the smoothing splines were scrutinized and intentionally kept as low as possible (1–2 inflection points) to minimize issues of overfitting the data.

GAM was also used to characterize the relationships between actual larval catch and simulated larval abundance. We used in situ Acanthurid abundances from trawl catches and compared them to modeled abundances from the forecast simulation parameterized after Zebrasoma flavescens. A key linkage between larval abundance and larval catch involves efficiency of the sampling gear (e.g., Clarke, 1983), thus the GAM was parameterized with variables that have been known to impact visual avoidance such as moon phase and moon angle. Additionally, moon phase is known to affect biomass and abundance of mesophotic boundary layer communities in the study region (Drazen, Forest & Domokos, 2011). Moon data was obtained from the open source astronomical package named PyEphem written in the programming language Python. Moon phase and moon angle were calculated for the individual trawls based on their position, date, and time of day/night. Because both the moon phase (amount of illumination) and moon angle (important scalar of the amount of illumination) could be independently or jointly important we entered them as a bivariate smoothing term in the GAM function. The k term for the smoothers were set to k = 5 for simulated larval abundance and k = 6 for the moon phase and moon angle smoothing term (to keep inflection points at a reasonable level), and a Poisson distribution was assumed for the GAM. While there are likely many other variables that could relate to the efficiency of the sampling gear, these were thought to be the most important because tow speed and weather conditions were relatively constant over the course of these surveys.

Results

Field sampling

A strong cyclonic eddie was present off the west shore of Big Island during field sampling. Additionally, there was a weaker, anti-cyclonic eddie near shore, moving water southward along the coast south of Kailua-Kona. Mean circulation on the Kohala shelf was north-eastward until the ‘Alenuihāhā channel (between Maui and Big Island) pushed surface waters westward (Fig. 3). Most of our stations were located in the smaller anti-cyclonic eddy near the Kona coast. Trawling took place between a new moon on June 30, 2011 and a full moon on July 14, 2011.

Figure 3 Current vectors during field sampling.

Mean HYCOM currents between July 2 and July13, 2011, for Big Island of Hawai‘i. Red circles show trawl sites during field sampling.

Larval shore fish counts were highest in stations further offshore (Fig. 4), consistent with the observations of Leis & Miller (1976) and Leis (1982). Offshore stations had the greatest diversity of shore fish, with individuals from all seven familial groups present (Table 1). Acanthurids were present at 8 of the 11 stations and mirrored total shorefish abundance patterns in the trawl catch (Fig. 4). Variability between replicates was low relative to variability between stations. The average length of the collected Acanthurids was 26.1 mm pre-caudal length (PCL); near their size at recruitment (Claisse et al., 2009).

Figure 4 Larval shore fish trawl catch at each site.

Summary of larval shorefish catch from Cobb trawl samples used in this study. Bar graphs show average counts of larval shore fish for each station caught during the SE1104 cruise. The graphs are placed in the location of the trawl.

Table 1 Trawl data from SE1104 cruise July 2011.

Summarizing larval shore fish counts from seven familial groups, as well as latitude, longitude and moon phase for all stations visited during SE1104 cruise from July1 to July 14th, 2011. Trawl 6 and 8 has no data on larval shore fish abundance because the cod end of the trawl was compromised by cookie cutter sharks (Isistius brasiliensis) and trawl catch lost.

Trawl number	Station number	Longitude	Latitude	Moon phase	Moon angle	Pomacanthidae	Acanthuridae	Chaetodontidae	Labridae & Cirrhitidae	Muraenidae	Fistulariidae	Mullidae	
1	1	203.50	19.65	4.5	36.8	24	5	2	3	1	2	15	
2	1	203.47	19.63	5.4	58.2	28	10	5	1	0	4	0	
3	2	203.87	19.67	10.4	25.0	14	1	1	1	1	0	0	
4	2	203.87	19.67	11.9	52.9	1	0	0	0	0	0	0	
5	3	203.80	19.98	18.4	12.7	2	0	0	0	0	0	0	
7	4	204.03	20.01	28.0	0.7	7	1	1	0	0	1	0	
9	5	203.85	19.30	38.9	0.0	1	7	5	0	0	1	0	
10	5	203.86	19.30	40.8	24.8	5	5	22	5	1	2	0	
11	6	203.49	19.35	50.4	0.0	5	4	4	4	2	2	17	
12	6	203.47	19.34	52.3	13.6	7	8	15	0	6	0	3	
13	7	203.54	19.50	61.8	0.0	3	6	12	1	4	4	6	
14	7	203.51	19.45	63.6	2.5	8	4	15	0	2	2	0	
15	8	204.17	19.33	72.9	0.0	4	2	0	1	0	0	0	
16	8	204.08	19.33	74.2	0.0	6	0	0	0	0	0	0	
17	9	203.98	19.53	82.0	0.0	4	0	0	0	1	0	0	
18	9	203.98	19.53	83.4	0.0	8	2	0	0	0	0	0	
19	10	203.81	19.86	89.8	0.0	1	0	0	0	0	0	0	
20	10	203.83	19.83	90.9	0.0	7	0	0	0	0	0	0	
21	11	203.64	19.99	95.5	0.0	1	0	0	1	0	0	0	
22	11	203.64	19.99	96.2	0.0	5	0	0	0	0	0	0	

Hindcasting simulation

A GAM was constructed to identify effects from vertical layering and PLD on the predicted origin by using these variables in a model predicting hindcast settlement success. PLD is highly significant (p < 0.001) for dispersal in the model. The PLD function shows that shorter PLD’s coincide with higher hindcast settlement success, and that after 40 days there is little change in the probability of originating from a suitable adult habitat (Fig. 5A). After 40 days of dispersal, the effect PLD has on dispersal ability remains stable, indicating that fish with PLDs longer than 40 days have the same potential for dispersal based on PLD alone. Thus, a 45 day PLD can be representative for longer PLDs, which has implications for future dispersal modeling as well as predicting dispersal ability based on PLD alone.

Figure 5 Effect of PLD and depth on modeled larval success.

Generalized additive model output showing relationships between modeled settlement success and (A) pelagic larval duration (PLD) and (B) depth strata used in biophysical model. Higher values denote higher modeled settlement success. Gray areas shows ± two standard errors.

The vertical function from the GAM suggests that the depth layer that is most optimal for larval shore fish in order to have originated from a suitable adult habitat is 100 m (p < 0.001) followed by the 50 m depth layer (p < 0.05) (Fig. 5B). There is no statistically significant difference between the effect of the surface strata or the two averaged (0–50 m and 0–100 m) strata on origin sites according to the GAM, indicating that the surface layer is disproportionally influencing the averaged depth scenarios.

Forecasting simulation

The forecast simulations showed that actual larval catch was significantly related to simulated larval abundance (p < 0.05) with the moon phase and moon angle surface (p < 0.001) explaining 67.5% of the deviance (Fig. 6). The relationship between the larval catch and simulated larval abundance was positive and the moon variables were incorporated with lower larval catch associated with higher moon angle (moon higher in the sky) and higher moon phase (brighter moon). There was indication of an intermediate moon phase corresponding to higher larval catch visualized by the hump on the surface in Fig. 6.

Figure 6 Relationship between larval catch and modeled abundance, and effect of moon on that catch.

Generalized additive model output showing relationship between trawled larval abundance of Acanthurids and (A) simulated larval Acanthurid abundance and (B) a moon phase and moon angle surface. The linear predictor is the observed and modeled abundance from figure (A). Higher values denote higher values of modeled larval catch and gray area in (A) shows ± two standard errors.

Discussion

We corroborated the larval dispersal model for July 2011 by field sampling, showing that the predicted larval densities are found to significantly feed into a plausible model of abundance. While treating larvae as passive particles in the dispersal simulation, and accounting for larval behavior and gear catchability through the lunar phase and angle variable in the GAM, we were able to account for nearly 68 percent of the variance when modeling Acanthurids. Furthermore, all seven familial categories of larval fish showed the same distribution pattern in trawl data with higher abundances off shore compared to near shore, lending further evidence of the presence of a common larval pool in the water off Kona (Sale, 1978).

There is ample evidence that larval fish undergo both ontogenic and diel vertical migration and that incorporating these behaviors into a biophysical model can increase the predictive power of the model (Drake, Edwards & Barth, 2011; Drake et al., 2013; Sponaugle et al., 2012; Sundelöf & Jonsson, 2012; Robins et al., 2013; Bidegain et al., 2013). However, little is known about the behavior of larval Acanthurids in general and Z. flavescens in particular. There are published swimming speeds for Acanthurids in the field and they illustrate a highly variable behavior and swimming speeds of late stage larvae (Sancho, Ma & Lobel, 1997; Leis & Carson-Ewart, 2000). Our model is parameterized for larvae approximately half way through their larval period (compared with in situ larvae that range between 10 and 30 mm in standard length). With such variable size and range of developmental stages, we could not venture a guess as to the swimming speed of the larvae, thus we opted for a conservative approach. Keeping the modeled larvae passive seemed appropriate in this initial evaluation of the performance of the model. Keeping larvae passive can also yield information about possible drivers of the observed distribution; achieving a high congruence between in situ and modeled distributions when modeled larvae were not attributed any behavior and were treated as passive drifters, indicates that the observed larval distribution is forced by physical oceanographic drivers. For example, Kough and colleagues (2013) identified an area in the Caribbean where lobster larvae aggregated, referred to as ‘nursery areas’ in the study, and hypothesized that this might be a physical oceanography-driven aggregation area for lobster larvae that could play an important role in the management of the lobster stock.

The presence of a mesoscale eddie during field sampling may have helped retain larvae near the trawl sites, driving the pattern of observed as well as modeled larval distributions. It has long been hypothesized that the eddies that form along the Kona coast are responsible for retaining larvae nearshore (Sale, 1970; Leis & Miller, 1976; Lobel, 1989; Lobel, 2011; Vaz et al., 2013), thus increasing he chance of settlement success and self-recruitment. Eddies form off Kona mainly during summer when the trade winds are stronger, which coincides with the major spawning season of most Hawaiian reef fish (Walsh, 1987; Bushnell, Claisse & Laidley, 2010). The eddies develop 10–15 days after a strong wind event and last for 55–70 days (Patzert, 1969; Lobel & Robinson, 1986). The majority of eddies move northwest at approximately 6 cm/s, which may aid in transporting larvae away from the Big Island and along the south perimeter of the Main Hawaiian Island chain (Patzert, 1969; Vaz et al., 2013). However, some eddies remain stationary in the lee of Big Island, and have been observed to remain in that location for 70 days (Lobel & Robinson, 1986), making it possible for larvae spawned on reefs along the leeward coast of the Big Island to be entrained in eddies and remain near the coast for the duration of their pelagic larval stage.

The larger ‘off shore’ eddie present during our field sampling (Fig. 3) spun up on or near June 1, 2011 and was present throughout sampling, whereas the smaller eddie located close to shore during field sampling spun up on or near July 3, 2011 and lasted the duration of field sampling. Considering that Z. flavescens spawns at full moon (Bushnell, Claisse & Laidley, 2010) and the PLD is estimated at 55 days (Claisse, McTee & Parrish, 2009; Christie et al., 2010), it is likely that larvae we caught during trawling were spawned near the full moon on either May 17 or June 16, 2011 and possibly entrained in the eddie shortly thereafter. Genetic analysis of the Acanthurid larvae can be done to verify the origin location of the larvae against published genetic connectivity data (Eble, Toonen & Bowen, 2009; Christie et al., 2010).

The hindcast simulation showed that larvae can reach the full extent of the Main Hawaiian Islands (MHI) in 40–45 days, which is important for future dispersal modeling studies. Connectivity patterns in the MHI will not change significantly if PLD is increased past 45 days, thus connectivity studies using a 45 day PLD can be representative of fish species with a longer PLD in Hawai‘i. The results from the hindcasting simulation suggest that the larvae should primarily occupy the 100 m depth layer if settlement success is the primary driver. Little is known about larval shore fish depth distributions around the Hawaiian Islands, however Acanthurid larvae have been found as deep as 575 m in the Caribbean, with the highest concentrations at or slightly above the chlorophyll a maximum (Oxenford, Fanning & Cowen, 2008) which in the study was at 100 m. Boehlert & Mundy (1996) reported that Acanthurids are found in the upper 80 m in waters off O‘ahu (mean mixed layer depth at Station ALOHA north of O‘ahu (22°45′N, 158°00′W) is 60.1 m (http://hahana.soest.hawaii.edu/hot/hot-dogs/mldepth.html)). The chlorophyll maximum at Station ALOHA is located near 110 m depth and the salinity maximum near 100 m (http://hahana.soest.hawaii.edu/hot/methods/chl.html; http://hahana.soest.hawaii.edu/hot/methods/salinity.html), making our hypothesis derived from model results that larvae should occupy the 100 m depth layer plausible.

By identifying both physical and biological features, such as preferred depth strata and the effect of PLD on dispersal ability, we can better understand and evaluate population dynamics in fish or invertebrate populations. Essential Fish Habitat (EFH) mapping is used in determining areas of high importance for a variety of commercially important fish species, but little to no consideration is taken to the larval stage of the fishes (D Kobayashi, pers. comm., 2012; Warner, Swearer & Caselle, 2000). Additionally, creating an archipelago-wide larval hotspot map will aid in identifying important larval fish habitat and this study lays the groundwork for that effort. The approach of generating a baseline population of organisms and applying a simulated capture process to allow comparison with actual catch data is similar to approaches used in stock assessment methodologies. The methodological approach called stock synthesis (Methot, 1990) is an example of such an inverse type approach (Murray-Smith, 2000). This approach is of value for identifying complex yet important drivers of the system under study when an observed value is the synergistic result of layered and interacting processes. In such instances, the parameters cannot be simply incorporated in conventional linear approaches modeling observed values against a suite of predictor variables. The insights into the capture process presented here might be useful for future inverse modeling of insular fish stocks and their recruitment dynamics.

As more detailed abundance, behavior and growth data become available for Zebrasoma flavescence, the modeling approach can be better tuned to model both patterns of connectivity and the spatiotemporal nature of the “larval pool.” Future trawl surveys can test the hypothesis of the 100 m depth occupancy, as well as target areas that are predicted to have high larval abundance if specimen acquisition is prioritized. Similarly, such areas of high larval abundance can potentially be protected from deleterious effects in an effort to preserve essential fish habitat and areas of particular concern, ideas that have received national attention in both the public and scientific communities (Warner, Swearer & Caselle, 2000; Kough, Paris & Butler, 2013). If species are recruitment-limited and/or the propagules are localized and vulnerable in time and space, pelagic areas might be warranted for management scrutiny.

Computer simulation modeling proved to be a powerful tool for furthering our understanding of larval transport in the Hawaiian Islands region. Even though the parameterization of the model was kept conservative with respect to larval behavior and life history traits (e.g., we did not include mortality, ontogenic or diel vertical migrations, or swimming behavior) we were able to reproduce in situ larval distributions and account for two thirds of the variance in the model. By ground-truthing the dispersal model with field sampling, we took steps towards making predictions about the spatiotemporal dynamics of the larval pool in west Hawai‘i. Future projects will incorporate additional years of in situ trawl catches so that we can make the biophysical model predictive, thus rendering it a valuable tool in management efforts of important commercial fish, not only the reef associated yellow tang (Zebrasoma flavescens) but for commercially important bottomfish in Hawai‘i such as eteline snappers.

Supplemental Information

Supplemental Information 1 Raw data for in situ and simulated acanthurid abundances, along with moon phase and moon angle for all trawls conducted used for input into forecasting GAM

Click here for additional data file.

Supplemental Information 2 Raw data of simulated larval settlement over five depths and five PLD’s at each trawl site from a hindcasting larval dispersal simulation. Also used for input data into hindcasting GAM

Depth strata refers to the five depth bins that we used, where depth strata 1 = surface layer, 2 = 50 m depth layer, 3 = 100 m depth layer, 4 = 0–50 m averaged, and 5 = 0–100 m averaged. Pelagic larval durations (PLD) are in days.

Click here for additional data file.

The authors would like to thank RJ Toonen for advice and support, B Mundy for invaluable help with larval fish identifications, Y Jia for HYCOM data, and the reviewers whose comments greatly improved this manuscript. We also thank the crew of NOAA Research Vessel Oscar Elton Sette and J Denton, A Shimada, N Shoji, M Seki, M Duncan, L Richards and E Howell for help with trawling and sorting.

Additional Information and Declarations

Competing Interests

Author Contributions

Data Availability

Donald R. Kobayashi is an employee of NOAA Fisheries, Pacific Islands Fisheries Science Center.

Johanna L. K. Wren conceived and designed the experiments, performed the experiments, analyzed the data, contributed reagents/materials/analysis tools, wrote the paper, prepared figures and/or tables, reviewed drafts of the paper.

Donald R. Kobayashi conceived and designed the experiments, performed the experiments, analyzed the data, contributed reagents/materials/analysis tools, wrote the paper, reviewed drafts of the paper.

The following information was supplied regarding data availability:

Raw data is available in the Supplemental Information.

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
