# Peer review of "Exploration of the “larval pool”: development and ground-truthing of a larval transport model off leeward Hawai‘i"

_PeerJ, doi:10.7717/peerj.1636_

## Round 0.1 · original submission · Major Revisions

· Academic Editor

Major Revisions

I am sure that you will consider all the suggestions made by the reviewers in order to improve your manuscript.

Reviewer 1 ·

Basic reporting

There is no ethics statement.

Experimental design

I am not sure I understand the study correctly. In the Introduction (line 90-91), the authors state that the study targets the yellow tang, Zebrasoma flavescens. However, in the following section (lines117-122), we read that identification at the species level for larval Acanthurids is not possible without genetic analyses (which were not done) and that the trawl data refer to the whole family of Acanthurids and not specifically for Zebrasoma. So which one is it? Is this a study on Zebrasoma or not?

Validity of the findings

I am not able to comment on the strength of the models presented here as this is not my area of expertise. That said, I find it very disappointing that the authors simulated larvae as 'passive' given the ample literature indicating that this is simply not the case. Although the authors acknowledge this point in their introduction, I feel the choice of a 'conservative approach' is a poor one as I am not really sure what this study is really adding to the field.

·

Basic reporting

The basic reporting was good. I have some syntax/editorial suggestions, but these are minor.

L33 – know:known

L37 – “Efficiently sampling in a vertically stratified sampling scheme is logistically intractable” – I don’t think that this is always necessarily true (i.e. is it true that you could never efficiently sample in a vertically stratified manner?). In this case, I agree it’s probably not feasible, but it should probably be qualified that way.

L88 – use:uses

General – watch the use of contractions (informal) – e.g. L120 – “It’s”, L234 “it’s”, L239 “doesn’t”, L244 “didn’t”

171 – sensus – This might be an improper pluralization? Maybe stay with sensu – “in the sense of”

179 – Citing R – the citiation makes it look like the Wood reference is the citation for R in general, when it is really just for the GAM package. It might be better to specify the package that was used and then cite it.

180 – Smoother functions… were used – which ones?

195 – I am probably not as up on my GAMS as I should be, but I’m not familiar with the “2-way surface term” terminology. As in… an interaction variable? I tried looking up “2-way surface term”, but was not able to come up with anything. Can the authors please clarify this?

244 “We kept simulated larvae passive and didn’t attribute them any behaviour” – awkward phrasing. Maybe something like – “ Larvae were not assigned any behaviour, and were considered to be passive drifters”?

269 groundtruted:groundtruthed

318 – italicize “in situ”

323 – The authors should probably provide a reference to the Deep-7 species – e.g. see http://www.fpir.noaa.gov/SFD/SFD_regs_5.html. Researchers outside of Hawaii might not be as familiar with this categorization.

Experimental design

147 – If the HYCOM data the authors were using has variable width resolution (and for all of 2011), then I assume they were using GLBa as opposed to GLBu (GLBu is a uniformly gridded product), and they may wish to make that distinction in the manuscript. Using GLBu may save them having to perform their own regridding if they intend to carry out further work using later date ranges.

182 – “Degrees of freedom were manually specified” – to ensure repeatability, what were the decision criteria?

154 – Polovina’s (and by extension, DeMartini’s) model takes into account turbulence at each time step using the term ε√(DΔt) (see Equation 2 on p135 of Polovina). When running the model backwards, was coalescence handled (i.e. reversing the diffusive process), and if so, how?

Validity of the findings

The findings seem justified to me. (No Comments)

Additional comments

I thought that the manuscript was well done, and although the review criteria state that decisions are not to be made on the basis of impact or novelty, to me the information was both novel and had impact. I was especially interested to see field data being compared with the modelled data, and I am of the opinion that more of these kinds of studies are badly needed. I am looking forward to seeing more of this type of work. If the authors have any questions regarding aspects of this review, they can feel free to contact me.

Reviewer 3 ·

Basic reporting

The study is an interesting investigation of larval transport mechanisms in the leeward of the Hawai'i island. The authors use both in-situ collected data and simulations to explore larval dispersal pathways, and estimate areas likely to originate settlers for reefs in the Hawaii island. The paper is generally well written, the methodology used, along with statistical analysis, were appropriate for the data collected and generated. Important implications can be drawn from this study, such as to inform local management on placement of protected areas. However, some shortcomings need to be addressed prior to publication.

Experimental design

Primarily, I would suggest you to review your objectives, making them clear in the introduction, adapting them to the data you have available and then re-focus your paper. Since the study is based on larvae collected one cruise, and the regional oceanography presents high variability, a study based on such a short time scale cannot elucidate mechanisms of transport (physical or biological) or generate a reliable larval pool for the region.

The paper is generally well-written, however in its present form it lacks focus. By re-focusing the manuscript based on clearly stated objectives, and available data, you would make it grounded, and enjoyable for reading.

I think that the cruise results are interesting and relevant, and thus should be explored. I believe the paper lacks a proper discussion of how the abundance of larvae sampled during the cruise relates to the biology of the species and to the circulation on that particular occasion.

If you consider the spawning period, frequency and fecundity of the yellow tang, you can explore questions that might arise, such as how the abundance of larvae collected is related to their spawning cycles?, could their abundance be different during other months and how fecundity would impact the modelling results? (Bushnell 2007; Bushnell et al. 2010). Also, include some analysis of the velocity field during the cruise period, and relate it to the presence of larvae. Important factors would be the presence of eddies, intensification of HLCC, etc, etc (there are numerous papers discussion larval transport in the leeward of the Hawaii Island and circulation, which can provide some insight for your manuscript, but are not used).

Validity of the findings

The intro and discussion rely heavily on speculation, much of it could be removed.

For example, on lines 263-269, "By combining larval fish data (e.g. abundance, temporal and geographic distributions and moon phase) with dispersal modeling we were able to develop a biophysical larval dispersal model that accurately predicts larval shore fish distributions along the west coast of Hawaiʻi Island," is a over-representation of the model capabilities: you were able to successfully represent one geographic region during a short time period of a specific year, and thus cannot extrapolate the results. This need to be clear on the text along the paper.

Additional comments

- Paragraphs 10 to 13 would read better following 22, and starting from Traditional maximum sustainable…

- 22: but also of larval dispersal and population connectivity

- 123: HYCOM data: not clear if the high resolution local implementation was used, or the global model. Description at parts is of high resolution local implementation (~0.04o degrees resolution, Jia et al. 2011), while it states that the model is a subset of the global HYCOM (which has a horizontal resolution of ~0.08o and it is data assimilative).

- 294-295: 100 m as the optimal layer: this layer presents lower velocities, expected to be conducive to retention (Vaz et al. 2013). However, is this a realistic conclusion considering the biology of the larva?

- 243: despite low ties, internal tides might be important to larval transport in the region

- Need to include that exogenous larval contributions are not considered, thus you can only provide settlers forecast for the "larval pool" from the reefs you are modeling

-308: typo

-310-312: Your main objective should be clearly stated in the intro. From the intro, I've understood that factors other than the physical oceanography would be investigated. I would expect at least one plot of the velocity field during the cruise duration, and some description of how it affects the abundance of larvae collected.

---

## Round 0.2 · accepted · Accept

· Academic Editor

Accept

The authors have improved the manuscript.

·

Basic reporting

OK

Experimental design

OK

Validity of the findings

OK

Additional comments

Looks good to me, nicely done. My only suggestion is that you don't need to add me to the acknowledgements. Reviewing articles is part of my job, and I'm of the opinion that acknowledgements are for people who go beyond the call of duty. Congratulations again, and Happy Holidays!